# Intra-Operative Video-Based Measurement of Biaxial Strains of the Ascending Thoracic Aorta

**DOI:** 10.3390/biomedicines9060670

**Published:** 2021-06-11

**Authors:** Shaiv Parikh, Berta Ganizada, Gijs Debeij, Ehsan Natour, Jos Maessen, Bart Spronck, Leon Schurgers, Tammo Delhaas, Wouter Huberts, Elham Bidar, Koen Reesink

**Affiliations:** 1Department of Biomedical Engineering, CARIM School for Cardiovascular Diseases, Maastricht University, 6229 ER Maastricht, The Netherlands; s.parikh@maastrichtuniversity.nl (S.P.); b.spronck@maastrichtuniversity.nl (B.S.); tammo.delhaas@maastrichtuniversity.nl (T.D.); wouter.huberts@maastrichtuniversity.nl (W.H.); 2Department of Cardiothoracic Surgery, Heart & Vascular Centre, Maastricht University Medical Centre, 6229 HX Maastricht, The Netherlands; berta.ganizada@mumc.nl (B.G.); gijs.debeij@mumc.nl (G.D.); ehsan.natour@mumc.nl (E.N.); j.g.maessen@mumc.nl (J.M.); elham.bidar@mumc.nl (E.B.); 3Department of Biochemistry, CARIM School for Cardiovascular Diseases, Maastricht University, 6229 ER Maastricht, The Netherlands; l.schurgers@maastrichtuniversity.nl

**Keywords:** aortic aneurysm, vascular biomechanics and mechanobiology, tissue deformation, image feature tracking

## Abstract

Local biaxial deformation measurements are essential for the in-depth investigation of tissue properties and remodeling of the ascending thoracic aorta, particularly in aneurysm formation. Current clinical imaging modalities pose limitations around the resolution and tracking of anatomical markers. We evaluated a new intra-operative video-based method to assess local biaxial strains of the ascending thoracic aorta. In 30 patients undergoing open-chest surgery, we obtained repeated biaxial strain measurements, at low- and high-pressure conditions. Precision was very acceptable, with coefficients of variation for biaxial strains remaining below 20%. With our four-marker arrangement, we were able to detect significant local differences in the longitudinal strain as well as in circumferential strain. Overall, the magnitude of strains we obtained (range: 0.02–0.05) was in line with previous reports using other modalities. The proposed method enables the assessment of local aortic biaxial strains and may enable new, clinically informed mechanistic studies using biomechanical modeling as well as mechanobiological profiling.

## 1. Introduction

Ascending thoracic aortic aneurysm formation is potentially lethal, with an annual incidence of approximately 6–10 cases/100,000 patient-years [1]. In current practice, the decision for surgical intervention considers the maximum diameter (≥5.5 cm) and/or growth rate (>1 cm/year) of the aneurysm [2,3,4,5]. However, treatment guidance based on these indicators still results in unexpected dissections in 60% of cases [4], as well as ruptures in 0.3% of cases for diameters < 4 cm and in 1.7% of cases for diameters < 5 cm [6]. In up to 95% of aortic dissections, aortic dimensions did not meet the guideline criteria prior to the event [7]. Additionally, recently introduced anatomical criteria such as aortic elongation have so far shown a limited predictive value [7]. Clearly, these current clinical metrics do not fully capture aortic wall integrity [4]. Aortic wall deformations as induced by the pulsating blood pressure could provide more insight into degenerative changes taking place in the aortic wall [4].

Many studies consider pulsatile changes in the aortic lumen area in response to changes in transmural pressure (i.e., distensibility) to estimate the structural stiffness of the aorta [8,9], and consider only circumferential stress [4]. However, these approaches for vessel stiffness and wall stress assessment ignore longitudinal deformations, which have been shown to be significant [10]. Moreover, accurate wall stress calculations in both healthy and aneurysmatic aorta are dependent on the biaxial mechanical behavior of the vessel wall, which is determined by the structural arrangement and properties of the extracellular matrix [11,12]. Taken together, the assessment of biaxial deformations appears highly important for understanding ascending thoracic aortic wall mechanics, material properties, and aneurysm formation.

Existing methods to capture the biaxial deformations of ascending aorta are based on non-invasive magnetic resonance (MR), computed tomography (CT), and ultrasound (US) imaging [8,9,10,13]. However, for the assessment of local biaxial mechanics of the ascending thoracic aorta, these imaging modalities are hampered by a more global region of interest definition (from the aortic root to brachiocephalic bifurcation) [8,9,10], as well as by a limited number of anatomical features that can be used to track deformations [9].

Considering the above, we developed an intra-operative video-tracking technique to assess local biaxial strains of the ascending thoracic aorta. Such an approach also enables investigation of the correlations between biaxial strain, tissue histology, as well as cell biology profiles in patients in whom tissue is resected [14,15]. Additionally, the strain measurements may serve as inputs to mathematical modeling studies [16]. In the present paper, we introduce and evaluate our method in patients undergoing open thorax surgery.

## 2. Materials and Methods

### 2.1. Study Population

Thirty-two consecutive patients undergoing open-chest cardiac surgery at Maastricht University Medical Centre were included. Prior to enrolment, patients gave written informed consent. The study was approved by the Maastricht University medical ethics committee (protocol METC2019-1235).

### 2.2. Video-Based Biaxial Strain Method Description

Figure 1 illustrates the key elements of the presented method. The biaxial strain method includes video recording, marker tracking, displacement measurements, filtering, and strain assessments, which are further detailed below.

#### 2.2.1. Intra-Operative Set-Up and Video Recording Settings

A camera (HERO7, GoPro Inc., San Mateo, CA, USA) with a mounting arm (GoPro stick, GoPro Inc., San Mateo, CA, USA) was secured to the surgical table at the head end (Figure 1A) during the routine clinical preparations for open-chest surgery. The arm consisted of a commercial action-cam extension arm (GoPro stick, GoPro Inc., San Mateo, CA, USA) and a custom mounting bracket adapted to the bed. The rigidity of the arm made sure that the camera position with respect to the opened chest was maintained during table tilting. To prevent manual artifacts during the starting and stopping of the recording, remote control was obtained by pairing a smartphone with the camera. The camera used was a commercial action cam (HERO7, GoPro Inc., San Mateo, CA, USA), with the following settings: 2.7k video resolution (screen resolution in pixels: 2704 × 1520; pixel size about ~0.08 mm), at 50 frames per second, and with a linear field of view.

After sternotomy and exposure of the ascending aortic region, four sterile surgical pledgets (BARD PTFE Felt Pledgets, Bard Peripheral Vascular, Inc., Tempe, AZ, USA) were sutured to the adventitial surface of the aorta (Figure 1B) cranial to the sinotubular junction. The two pledgets of a pair were placed as diametrically opposite as possible (Figure 1B). For all cases, the (axial) distance between the two pairs of markers was kept between 0.5 to 1 times the diametrical distance, as exemplified in Figure 1B.

Just prior to recordings, the camera was manually positioned to ensure that its focal plane was parallel to the imaging plane as defined by the four markers. The distance between the camera and imaging plane was about 40 cm (Figure 1A). For calibration, a sterile steel or paper ruler was placed in the imaging plane.

#### 2.2.2. Blood Pressure Conditions and Recording Protocol

Systolic and diastolic blood pressure (BP) were measured using a regular arterial pressure catheter (Edward Lifesciences, Irvine, CA, USA) placed in the (left) radial artery. Subsequently, pressure catheter readings were taken from the clinical routine hemodynamics monitor (stored in a case report form) just prior to and directly after video recordings. Videos were made during both ‘low’ and ‘high’ pressure conditions that were induced by table-tilting (Figure 1C).

The first video recording was obtained with the patient in either supine (neutral), anti-Trendelenburg, or Trendelenburg position. The second video recording was then obtained at a different pressure level after tilting the table. The initial table position determined whether the second measurement was made with an increased or decreased BP condition, utilizing the expected BP differences between positioning: Trendelenburg BP > supine BP > anti-Trendelenburg BP. This means that a low-pressure condition was induced by either a neutral or anti-Trendelenburg position, with a corresponding high-pressure condition achieved by Trendelenburg or neutral position, respectively (Figure 1C). The sequence (low-then-high or high-then-low) was arbitrary and determined by the surgeon, who merely targeted a mean arterial pressure (MAP) difference of about 10 mmHg between positions, for which positioning sequence was not relevant. It should be noted that the aim was to measure strains at a distinctly different transmural pressure without invoking pharmacological side-effects.

Video recording duration was 12 s to capture at least two ventilation cycles, enabling assessment of (ventilation) cycle-to-cycle reproducibility. The video recordings and pressure readings did not span more than 2 min, after which the surgical procedure continued.

In the aneurysm-repair cases, the resected tissue was further sectioned and stored in vials for genetic and tissue characterization. These data are not included in the analysis as they are beyond the scope of the present paper. After the procedure, all video files (.mp4 format) and blood-pressure case report forms were stored on a secure local server for further processing.

#### 2.2.3. Video Processing and Strain Calculations

To determine diameters and axial lengths from acquired videos, markers on the ascending aorta were tracked across all video frames (Figure 1B), using a proprietary program written in MATLAB (MATLAB R2018a; The MathWorks, Natick, MA, USA). In the first step, the region of aorta with markers and ruler from the first frame was selected manually using the MATLAB Image Processing Toolbox function imrect. The same imrect box was used to crop all other frames in the video. Subsequently, small regions around the markers were manually assigned to obtain sub-images used for correlation-based tracking across the video frames. A normalized cross-correlation matrix was used and implemented by the function normxcorr2. Sub-image displacements between frames were defined as those at which the normalized cross-correlation coefficient was maximum. Using the obtained positions of the markers in all video frames, dimensions of the line segments connecting the markers were determined as a function of time, yielding diameter (*D*) and axial length (*L*) signals (illustrated in Figure 1D,F).

To calibrate the measured distances to physical distances (in mm), a known reference distance visible on the ruler in a video frame was used. Note that this calibration is relevant only for the diastolic diameter (*D*_d_) and length (*L*_d_) estimates but not for the strain estimates (see calculations below).

The *D* and *L* signals were subjected to Fourier analysis to identify the lowest frequency (about 0.2 Hz in our subjects) resulting from the ventilation cycle (Figure 1D,F). Ventilation cycles and individual beats were then segmented manually from the raw *D* and *L* signals, with the start and end points selected manually. Subsequently, signals were smoothed, using a zero-phase, 6-point, bi-directional (forward and backward), moving-average filter (filtfilt, MATLAB). Beat-by-beat minimum and maximum points were determined, yielding diastolic values (*D*_d_ and *L*_d_) and the corresponding systolic-diastolic changes (Δ*D* and Δ*L*). The circumferential strains and axial strains were then calculated for each beat as the engineering strains Δ*D*/*D*_d_ and Δ*L*/*L*_d_, respectively (Figure 1E,G). The diastolic dimensions and strains were averaged over the number of beats (in one full ventilation cycle) by taking the median.

In addition, pulse pressure (Δ*P*) and circumferential distensibility (Δ*D*/(Δ*P·D*_d_)) were calculated. The two ventilatory cycles (Figure 1D,F) present in every recording were considered as repeated measurements (*m* = 2), to assess reproducibility (see Statistical analysis).

### 2.3. Statistical Analysis

Data are presented as median [25th, 75th percentile], unless noted otherwise. Non-normality of presented measurement data was verified using a Shapiro–Wilk test. For *p* < 0.05, the distribution of measurements was not considered normal.

We tested for statistical differences between low- and high-pressure conditions, and for differences between cranial and caudal strain as well as between lateral and medial strain measurements using paired Wilcoxon Signed-Rank tests.

Based on the two ventilatory cycles in each recording (see above), the intra-measurement standard deviation (σ_intra_) was determined as the square root of the average of variances of the repeated measurements for each recording:(1)$$\sigma_{\mathrm{intra}}=\sqrt{\frac{1}{n}\overset{n}{\underset{i=1}{\sum }}s_{i}^{2}}$$
where, $\frac{1}{n}\overset{n}{\underset{i=1}{\sum }}s_{i}^{2}$ is the average of variances for *n* subjects. We expressed reproducibility using the coefficient of variation (CV), defined as $\sigma_{\mathrm{intra}}$ divided by the corresponding sample mean value $\bar{x}$ of each group times 100%:(2)$$\mathrm{CV}=\frac{\sigma_{\mathrm{intra}}}{\bar{x}}\cdot 100\%$$

Using Bland’s estimate of the uncertainty about reproducibility, the $\sigma_{\mathrm{intra}}$ we calculated has an uncertainty of 25%, given the number of repetitions *m* = 2 and *n* = 30 [17].

## 3. Results

Twenty-seven male and five female patients were included in the present analysis. Videos of patients where tracking of markers was not accomplished—either due to de-correlation (*n* = 1) or loss of visibility of the tracking marker in all frames (*n* = 1)—were excluded. Patient characteristics are mentioned in Table 1. The age of the patient population was 64 ± 12 years (mean ± SD).

Table 2 shows the blood pressure (BP) and video-derived measurements. In one subject we did not obtain video results in the high-pressure condition, due to loss of view on one marker (hence *n* = 29 for high pressure). There were two subjects in which the ruler appeared not visible enough. In these cases, marker dimensions (4.8 mm × 6.0 mm) were used for calibration instead of the ruler.

Diastolic and systolic blood pressures (DBP and SBP) changed significantly (*p* < 0.001) from low- to high-pressure conditions (Table 2), with a mean arterial pressure (MAP) difference of 10 mmHg between both conditions. The absolute dimensions (*D*_d_ and *L*_d_) and strains (Δ*D*/*D*_d_ for caudal and cranial, and Δ*L*/*L*_d_ for lateral) did not vary significantly with the change in transmural pressure. Medial axial strain increased significantly by 0.01 from low- to high-pressure (*p* = 0.007). For most direct measures, the measurement variability was of the order 10%, although for the strains, the CVs tended to be higher but remained below 20% (Table 2).

From low- to high-pressure, pulse pressure (Δ*P*) increased by 8 mmHg (*p* < 0.001) and distensibility showed a trend towards decrease: from 7 MPa^−1^ to 4 MPa^−1^, *p* = 0.078 for caudal and from 6 MPa^−1^ to 3 MPa^−1^, *p* = 0.020 for cranial.

Figure 2 summarizes differences between locations. For the high-pressure condition, we detected differences between caudal and cranial circumferential strains (0.03 and 0.02, respectively; *p* = 0.005), as well as between medial and lateral axial strains (0.05 and 0.04, respectively; *p* = 0.006). Interestingly, for the low-pressure condition, we did not observe clear differences between locations.

## 4. Discussion

At present, the assessment of aortic aneurysm progression is limited to monitoring (slow changes in) vessel dimensions [2,3,7]. Current clinical imaging modalities show limitations for assessing dynamic biaxial strains in the ascending thoracic aorta. In the present study, we introduce and evaluate an intra-operative video-based method for measuring biaxial strains in the ascending thoracic aorta. Our findings show the ability–in patients with and without aneurysm–to capture local circumferential as well as axial strains, including changes in response to changes in transmural pressure, with very acceptable reproducibility (CV range 8–19%).

Our approach differs from existing methods reported in previous studies [8,10], where global rather than local longitudinal deformations were quantified. These global approaches consider as a region of interest the entire trajectory between the aortic root at the one end and the brachiocephalic bifurcation at the other end. Our method enables the measurement of local axial strains, and particularly in aneurysmatic regions of the ascending aorta.

Longitudinal strains obtained using global approaches are in the order of 6–9% [8,10] in patients and in the order of 15% in presumably healthy volunteers [10], while our study subjects exhibited local longitudinal strains in the order of 4–5%. The global approaches idealize the entire region of interest for longitudinal strains into a cylindrical geometry, presuming a homogenous distribution of strains [8,10]. However, owing to the bend in the aorta, possible differences in local strains between inner and outer curvature may exist. Our method (with four markers) does enable the assessment of such differences (Figure 2). We expect that the bent form with its complex deformation, even if wall material were assumed homogenous, may lead to differences between the medial and lateral axial strains. Yet, at this point, we do not have a clear concept of how to interpret such differences in local longitudinal strains. Changes in such differences could be instrumental in identifying remodeling processes at the constitutive level.

Circumferential strains reported in the literature [8,9,10,13,18] range between 3 and 14%, whereas we observed circumferential strains in the order of 2–3%. The strains we found are in the same range as those reported by Morrison et al. [9], who observed substantially decreased circumferential strain at the aortic root and the ascending segment (proximal to the first branch of the brachiocephalic trunk) in older patients (3% strain; mean age of 68 years), as compared to younger patients (10% strain, mean age of 41 years) using cardiac-gated CT image data. Van Disseldorp [18] also estimated ascending aortic circumferential strains in the order of 4–7% in aneurysm patients (age ranging from 44 to 72 years) using 3D transesophageal (TEE) ultrasound. On the other hand, Wittek et al. [13] reported circumferential strains in the order of 11% for healthy young volunteers (median age of 25.5 years) using 3D ultrasound speckle tracking, which inclined more towards the circumferential strain range of 7–14% [8,10] reported by global approaches using magnetic resonance imaging (MRI). In addition, the pressure-dependence of strain measurements must also be considered, as likely caused by the non-linear elastic behavior of the blood vessel wall [16,19]. Our observation of a decrease in distensibility with increasing pressure (Table 1) corroborates this clinically important phenomenon. Taken together, differences in the magnitude of strains may well be attributed to differences between studies pertaining to age [20], disease stage of the study population, imaging modality, as well as hemodynamic conditions.

When considered in more detail, strain differences between our novel method and existing imaging modalities may also be due to differences in temporal and spatial resolution. 3D ultrasound enables imaging of the aorta at a moderate temporal frequency of 11–25 Hz [13] with a voxel size on the order of 0.2–0.7 mm [13,18]. However, access to the whole segment of the aorta is cumbersome, thereby limiting the ability to (keep) focus on the desired region of interest. Additionally, MR and CT rely on the reconstruction of the cardiac cycle from multiple beat snapshots (30 phases per cardiac cycle [8,10]; spatial resolution: 0.5–1.5 mm [8,10]), presuming that consecutive beats are ‘identical’. In contrast, our method is able to capture single beats at a 50 Hz frame rate as well as at 10 times better spatial resolution: about 0.08 mm.

Some methods with established imaging modalities use a fixed location in the image to determine circumferential strains [8,10]. Such an approach may lead to errors due to through-plane movement of the aorta, yielding an overestimation in the circumferential strain of up to 50% as reported by Morrison et al. [9]. Identification of fixed anatomical markers in the images avoids this problem, but these are generally limited to bifurcations or valve features. Our approach solves the lack of anatomical markers for a local region of interest by utilizing pledget markers placed at desired locations.

The local differences (between caudal and cranial) in circumferential strain were found to suggest that our method may also detect these regional differences (Figure 2). However, we did not expect to find a trend towards lower cranial strains when compared to the caudal location (Figure 2). Clearly, the relevance and potential of regional differences in tissue properties due to local strain variations (owing to the complex geometry of the aorta) call for future, sufficiently powered studies. In that respect, it is highly relevant to consider the reproducibility of our strain-measurement method. Overall, CVs were less than 20%, and these reproducibility estimates were obtained with an uncertainty of about 25% (see Statistical Analysis). Assuming an expected absolute difference in strain of about 0.01 (e.g., a difference of 0.04 in one group and 0.03) and the 20% reproducibility as the lower limit of variability, the required minimum sample size for a case–control study would be about 11 per group (groups of equal size, power 1−β=0.8, type-I error α=0.05).

We deem our method particularly useful for mechanistic studies. For instance, our method may be of interest to investigate the correlation between CT-based atherosclerotic burden (i.e., regional wall thickness abnormalities) and local biaxial strains [21]. Such studies would be very interesting, because the (assumed) changes in stiffness with atherosclerotic burden would depend on, e.g., how and to what extent the intimal disease processes affect medial structure and properties. The ‘innate’ invasive nature of our method obviously precludes a wider clinical application.

For computational (e.g., finite-element) studies, the availability of high-quality input data is essential [22]. Constitutive models, that capture the elastic properties of the aortic wall under large deformations, rely on reliable estimates of (local) strains and biaxial data. Direct mapping of video marker displacements onto finite element models, using an inverse modeling approach, may enable the estimation of constitutive properties. Besides facilitating realistic computational modeling, well-parameterized constitutive models may provide insight into microstructural changes such as increased cross-linking of collagen and/or elastin degradation, which in turn may help in developing therapeutic approaches in silico [22,23]. Furthermore, computational modeling estimates based on our strain measurements may serve as a benchmark for non-invasive imaging techniques. Moreover, comparative studies correlating our method to MRI or TEE could provide the knowledge base to translate mechanistic insights towards clinical use.

Lastly, reliable strain data accompanied by immunohistochemical and cytochemical assays may allow comprehensive studies on the mechanobiological factors in ascending thoracic aortic aneurysm formation [14,15].

## 5. Conclusions

The proposed intra-operative, video-based method enables the assessment of regional biaxial strains of the ascending thoracic aorta with very acceptable precision. Our method provides a steppingstone towards clinically informed mechanistic studies using biomechanical modeling as well as mechanobiological profiling.

## Figures and Tables

**Figure 1 biomedicines-09-00670-f001:**
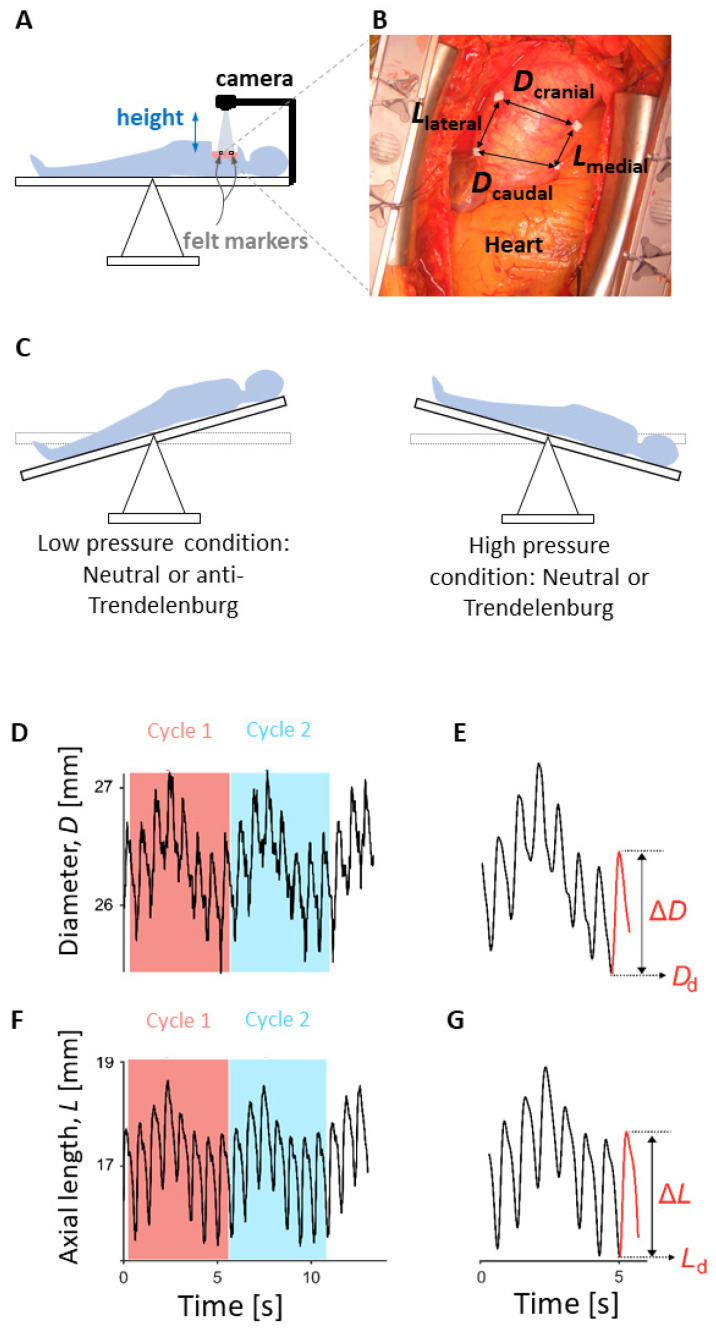
Key elements of the intra-operative, video-based strain measurement method. (**A**): Intra-operative set-up. The remote-controlled camera mounted on a rigid arm attached to the table was positioned at a height of about 40 cm above the imaging plane, as defined by the markers, with a viewing angle close to zero degrees. (**B**): Single video image showing heart on bottom and markers sutured on the adventitia of an ascending aortic aneurysm. Arrows between markers define the positions at which cranial and caudal (circumferential) strains, and medial and lateral (axial) strains were assessed. (**C**): Low and high transmural pressure conditions were created by tilting the table. Tilted table position for low-pressure condition shown is anti-Trendelenburg, while tilted-table position for high-pressure condition shown is Trendelenburg position. Horizontal position of the table is referred to as neutral position. (**D**,**F**): Examples of diameter and axial length signals, comprising two ventilatory cycles. (**E**,**G**): Smoothed signals for diameter and axial length, respectively (single ventilation cycles). The variables *D*_d_ and *L*_d_ represent magnitudes of diameter and axial length corresponding to diastolic pressure, while Δ*D* and Δ*L* are deformations of diameter and axial length caused by the change in pressure from diastole to systole (i.e., pulse pressure).

**Figure 2 biomedicines-09-00670-f002:**
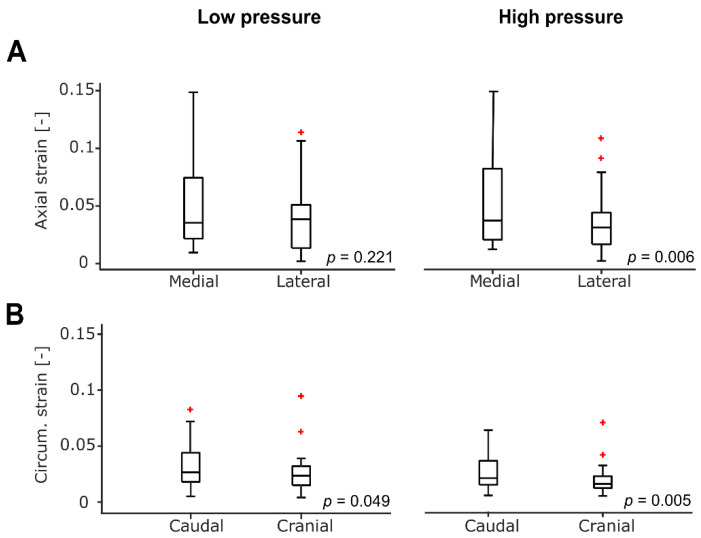
Potential for detecting local strain differences. *p*-values indicate paired Wilcoxon Signed Ranks testing, with *n* = 30 for low and *n* = 29 for high pressure. Boxplots indicate medians [25th, 75th percentile], with whiskers indicating variability beyond the first and the third quartile, while red plus signs are the outliers. (**A**): Lateral axial strains tended to be lower than those captured medially, but only achieving statistical significance in the high-pressure condition. (**B**): Cranial circumferential strains tended to be significantly lower than at the caudal location. Note: one outlier not shown in B (values for caudal and cranial strains > 0.2 for both pressure conditions).

**Table 1 biomedicines-09-00670-t001:** Patient characteristics.

Type of Surgical Intervention	Aortic Repair	AVR	CABG
Number of patients	17	9	6
Male/female	14/3	7/2	6/-
Age (years)	62 ± 9	59 ± 17	72 ± 5

AVR = aortic valve replacement; CABG = coronary artery bypass grafting; age is presented as mean ± SD.

**Table 2 biomedicines-09-00670-t002:** Blood pressure and video-derived measures.

	Pressure Conditions				CV (%)	
Measurements	Low (*n* = 30)	High (*n* = 29)	*p* Value * (-)		Low	High
**Blood pressure (mmHg)**						
SBP	84 [70, 93]	98 [90, 105]	<0.001		4	3
DBP	47 [36, 52]	54 [47, 62]	<0.001		10	5
Δ*P*	35 [26, 48]	43 [33, 53]	<0.001		4	7
**Diastolic diameter, *D*_d_ (mm)**						
Caudal	30 [25, 36]	30 [23, 38]	0.97		2	2
Cranial	33 [23, 40]	34 [23, 41]	0.8		2	2
**Circumferential strain, Δ*D*/*D*_d_ (-)**						
Caudal	0.03 [0.02, 0.05]	0.03 [0.02, 0.05]	0.19		8	8
Cranial	0.02 [0.01, 0.03]	0.02 [0.01, 0.03]	0.3		19	12
**Distensibility (MPa^−1^)**						
Caudal	7 [4, 9]	4 [3, 9]	0.078		8	13
Cranial	6 [3, 7]	3 [2, 6]	0.02		22	15
**Axial diastolic length, *L*_d_ (mm)**						
Medial	20 [17, 27]	19 [17, 25]	0.92		2	3
Lateral	24 [20, 30]	24 [19, 29]	0.97		2	1
**Axial strain, Δ*L*/*L*_d_ (-)**						
Medial	0.04 [0.02, 0.08]	0.05 [0.02, 0.09]	0.007		10	11
Lateral	0.04 [0.01, 0.05]	0.04 [0.02, 0.06]	0.3		19	12

Values are indicated as median [25th, 75th percentile]; CV = coefficient of variation (($\sigma_{\mathrm{intra}}$/mean)·100%); SBP = systolic blood pressure; DBP = diastolic blood pressure; Δ*P* = pulse pressure; Distensibility = (Δ*D*/(Δ*P*·*D*_d_·133))·10^6^; * values compared for low pressure vs. high pressure conditions using paired Wilcoxon Signed-Rank test (performed on 29 subjects); *n* = number of subjects.

## Data Availability

Data are available upon reasonable request addressed to the corresponding author.

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
