# Peer review of "Intra-Operative Video-Based Measurement of Biaxial Strains of the Ascending Thoracic Aorta"

_biomedicines, 2021, doi:10.3390/biomedicines9060670_

Round 1

Reviewer 1 Report

I read with interest the manuscript “Intra-Operative Video-Based Measurement of Biaxial Strains of the Ascending Thoracic Aorta”. The authors propose a video-based measurement of aortic deformation.

            My observations:

  1. Aortic size should be taken into account as a confounding variable in the strain measurements, as it directly affects wall tension
  2. Age of the patients is another factor which complicates the interpretation of their results. While younger patients have an aorta that is more elastic, older ones have conceivably a higher atherosclerotic burden which translates into a more rigid vessel wall.
  3. Age also affects aortic size, as the size of the aorta increases with age
  4. The preoperative CT scan which would assess atherosclerotic burden would be useful in identifying aortas with reduced elasticity.
  5. Direct measurements of aortic deformation, as proposed in the paper, are useful in the experimental setting, but their interpretation must take into account all variables that might affect results

Reviewer 2 Report

Intra-operative video-based measurement of biaxial strains of the ascending thoracic aorta

Dear Editor, thank you for the opportunity to review this article.

The authors do a great job. The aim of the study is to test the accuracy in measuring the local biaxial strains of the ascending thoracic aorta with a new method based on an inytraoperative video-tracking technique. The ascending aorta of 32 patients, during open heart surgery, has been analyzed (27 male and 5 female). Not all the measurements were completed. Additionally, when resected, the aortic wall was histologically studied, even though the results were not included in discussion. The article is very well written and structured.

My concerns relate to the clinical usefulness of this work.

For this reason I ask the following questions:

1) why did the authors not compare their strain variations with those observed on TEE during surgery time, or with those measured on preoperative MRI?

2) How do they think their method can have clinical significance?

suggestion:

1) EACTS / AHA / AATS guidelines for the treatment of ascending aortic aneurysms should be included in the introduction.

Reviewer 3 Report

In a study by Parikh et al., the authors developed an intra-operative video-tracking technique to 56 assess local biaxial strains of the ascending thoracic aorta and aimed to evaluate their method in patients undergoing open thorax surgery. The enrolled 32 consecutive pts scheduled for open heart surgery. The use of MATLAB application constantly increases in medical area which is promising. What is missing for me, what is a practical advantage of a computational study performed by authors? Do their findings impact somehow clinical outcomes? It would be useful, in my opinion, to extend discussion in this matter, or maybe present some glimpse of it in the results section?

Round 2

Reviewer 1 Report

I read with interest your response to the issues I raised and consider that they have been addressed in your revised manuscript. 

Reviewer 2 Report

The authors satisfied my questions.